# Health selection on self-rated health and the healthy migrant effect: Baseline and 1-year results from the health of Philippine Emigrants Study

**Adrian Matias Bacong**[1]*, **Anna K. Hing**[2], **Brittany Morey**[3], **Catherine M. Crespi**[1], **Maria Midea Kabamalan**[4], **Nanette R. Lee**[5], **May C. Wang**[1], **A. B. de Castro**[6], **Gilbert C. Gee**[1]

**1** University of California-Los Angeles Fielding School of Public Health, Los Angeles, California, United States of America, **2** University of Minnesota – Twin Cities, Minneapolis, Minnesota, United States of America, **3** University of California-Irvine, Irvine, California, United States of America, **4** University of the Philippines Population Institute, Quezon City, Philippines, **5** USC-Office of Population Studies Foundation, Inc., University of San Carlos, Cebu City, Philippines, **6** University of Washington, Seattle, Washington, United States of America

* adrianbacong@ucla.edu

**Data Availability Statement:** The Health of Philippine Emigrants Study (HoPES) data are available to researchers via a "Data Use

## Abstract

Studies of migration and health focus on a "healthy migrant effect" whereby migrants are healthier than individuals not migrating. Health selection remains the popular explanation of this phenomenon. However, studies are mixed on whether selection occurs and typically examine migrants post-departure. This study used a novel pre-migration dataset to identify which health and social domains differ between migrants and their non-migrant counterparts and their contribution to explaining variance in self-rated health by migrant status at pre-migration and 1-year later. Data were used from the baseline and 1-year follow-up of the Health of Philippine Emigrants Study (HoPES). We used multivariable ordinary least squares regression to examine differences in self-rated health between migrants to the U.S. and a comparable group of non-migrants at baseline (premigration) and one year later, accounting for seven domains: physical health, mental health, health behavior, demographics, socioeconomic factors and healthcare utilization, psychosocial factors, and social desirability. A migrant advantage was present for self-rated health at baseline and 1-year. Accounting for all domains, migrants reported better self-rated health compared to non-migrants both at baseline (β = 0.32; 95% CI = 0.22, 0.43) and at 1-year (β = 0.28; 95% CI = 0.10, 0.46). Migrant status, health behavior, and mental health accounted for most of the variance in self-rated health both at baseline and 1-year follow-up. This analysis provides evidence of migrant health selection and nuanced understanding to what is being captured by self-rated health in studies of migrant health that should be considered in future research.

Agreement", which helps to protect the identities of participants and ensures data integrity. Our entire migrant sample consists of newly arrived immigrants. Immigration is a contentious issue within the United States and there have been some political entities that have sought to revoke visas for immigrants. Therefore, our data are accessible, but protected via a screening process to ensure that users are seeking the data for research purposes only. These restrictions are recommended by standards from the University of California Los Angeles, the University of San Carlos - Office of Population Studies Foundation, Incorporated Philippines, and the Ethics Review Committee of the University of San Carlos, Philippines. To submit a "Data Use Agreement", please email hopesresearchstudy@gmail.com.

**Funding:** This work was supported by funding from the U.S. National Institutes of Health (1R01HD083574-01A1, R01MD012755 and 1R21CA137297) to Gilbert C. Gee and A.B. de Castro. The Health of Philippine Emigrants Study (HoPES) also received general support from the California Center for Population Research at UCLA (CCPR), which receives research infrastructure funding (P2C-HD041022) from the Eunice Kennedy Shriver National Institute of Child Health and Human Development (NICHD). HoPES also received support from the UCLA Clinical and Translational Science Institute (CTSI) which receives grant support from the NIH National Center for Advancing Translational Science (NCATS) (UL1TR001881). Adrian Matias Bacong was also supported by the 2019 University of California Los Angeles Graduate Research Mentorship Fellowship and the National Institute on Minority Health and Health Disparities of the National Institutes of Health (F31MD015931). The content is solely the responsibility of the authors and does not necessarily represent the official views of the National Institutes of Health. The funders had no role in the study design, data collection and analysis, decision to publish, or preparation of the manuscript.

**Competing interests:** The authors have declared that no competing interests exist.

# Introduction

Migrants often report better self-rated health (SRH) compared to those who do not leave the sending country (henceforth "non-migrants") [1, 2]. This may result from "health selection"—the idea that healthy people are more apt to migrate than sick people [3]. However, it is not clear which aspects of physical, mental, and social health drive this pattern [4–6]. The goal of this paper is to examine the relative contributions of factors such as physical health, mental health, social capital, and socioeconomic status to differences in SRH between migrants and non-migrants in the Philippines to the United States (U.S.) migration context.

## Self-rated health, immigration, and health selection

SRH is a particularly useful measure among international and immigrant populations, as it can be a cost-effective indicator of physical, mental, and social health, and mortality [7–10]. In general and for most sending countries, migrants typically report better SRH compared to non-migrants [1, 2]. A comprehensive study using data from the World Health Survey and American Community Survey found evidence of health selection on SRH between migrants and non-migrants in 18 of the top 19 sending countries to the United States (U.S.) [1]. While informative, these studies examined a limited set of covariates that could contribute to differences in SRH. In particular, Ro, Fleischer, and Blebu [1] limited their analysis of health selection to country level covariates (e.g. gross domestic product, infant mortality rate, geographic and cultural distance) as possible explanations for health selection. This ecological analysis is useful in understanding selection at the country level. However, this study does not answer the question as to what SRH actually captures when comparing migrants to non-migrants. Does SRH capture differences in demographics, physical health, social health, or something else? Few studies capture the myriad of different factors that contribute to SRH [3, 11–13]. Those that have attempted to examine the contribution of these factors have been limited to demographic and socioeconomic factors [3, 12] and fewer have examined health related factors [13]. Even so, these factors are treated as simple covariates, rather than attempting to quantify the contribution of these variables in explaining differences.

## Understanding migrant health selection from a social determinants of health framework

A social determinants of health (SDoH) perspective can be a useful framework for understanding the impacts of migration on health selection and health longitudinally. The SDoH framework emphasizes the interplay of social and environmental factors on the biological expression of health [14]. Multiple factors have been identified as SDoH including demographic factors, educational access and healthcare access, social cand community context, and economic stability [14]. Immigration has been identified as a social determinant of health that both facilitates and restricts immigrants' access to healthcare resources and affects their overall wellbeing in multiple domains [15]. In this paper, we consider the intersections of immigration as a social determinant of health with self-rated health, a global expression of overall health that encompasses different domains of health and wellbeing [16, 17]. We focus on one aspect of this intersection, health selection by migration status, which has been a major point of debate in the immigration literature [1, 3, 5, 18–21].

Health selection may result in part from differences in demographic factors between people who migrate and those who do not, such as age, gender, and English language proficiency [1, 5, 12, 22, 23]. English language proficiency is of special importance, as interpretation of one's

SRH can depend on interpretation and language. Different cultural groups may have distinct interpretations of the response categories of SRH, especially when translated [1, 3, 12].

As SRH is often used as an indicator of overall physical, mental, and social health, we expect that measures consistent with the concept of allostatic load—a cumulative measure of physiological wear and tear of the body in response to stress [24], emotional distress, perceived stress, cognitive functioning, and health behaviors might explain differences in SRH between migrants and non-migrants. Previous studies have noted how physical health (in the form of height or the number of health conditions) may differ between migrants and non-migrants [5, 6, 13].

Socioeconomic advantages (e.g. higher education, more fiscal resources) among migrants relative to non-migrants may contribute to health selection [6, 22]. Moreover, having access to healthcare may allow for better health in general [25, 26]. Additionally, social factors are important components of the migration experience that may have consequences on health [5]. For example, social capital can also function as a resource to better support migrants both before and after migration [5] by increasing access to health promoting resources such as healthcare and socioeconomic [27]. However, social capital may decrease after migration since social ties may be broken and new ties may be established.

Additionally, social isolation resulting from migration may contribute to worse health due to the inability to integrate in a new society [28]. Finally, health selection on migrant status may simply be an artifact of social desirability, such that when asked about their health status, migrants may respond in a more socially desirable manner than non-migrants to inflate SRH measures and affect immigration admissibility [29].

## Migration in the Philippines context

Today, the majority of authorized U.S. migration today comes from Asian countries; about 37% of all authorized immigrants were from Asian countries [30]. However, when studying health selection, many of the seminal studies out of the United States (U.S.) have compared migrants and non-migrants from Mexico.

These initial studies have focused on cross-sectional analyses. Using cross-sectional analyses limit the field's ability to examine if health changes over time and what factors contribute to change. Finally, because data are often collected years after migrants arrive in their host country, conclusions on whether health advantages existed prior to migration are difficult to determine.

The Mexican Family Life Survey (MxFLS) is the one major international study that explicitly addressed the limitations of cross sectional studies, but found mixed evidence of a healthy migrant effect [11, 22]. Some claim that these mixed findings may be due to Mexico having a lower barrier to migration than other countries because of its shared land border with the U.S. The lower barriers to migration could contribute to less stress during the migration process, specifically among authorized migrants (i.e., migrants with eligible visas) [3]. Moreover, these studies examined a limited set of factors that migrants could select upon (e.g., demographic and socioeconomic factors). This trend seen among Mexican migrants has also been corroborated by other cross-country analyses. Another study examining health selection of migrants from 19 major sending countries to the U.S. found evidence of negative health selection among Mexican migrants only [1]. It is worth noting that migration from Mexico to the U.S. is not representative of all migration. Mexicans made up about 15% of authorized immigration to the U.S. in 2016 [30]. Thus, while migration from Mexico is important to consider in studying migration and creating migration policy, it is clearly not representative of all migration to the U.S.

Longitudinal studies utilizing the MxFLS continue to find mixed evidence of selection and a healthy migrant effect. One study has noted that Mexican migrants report both improvements and declines in health compared to their non-migrant counterparts two years after migration [13]. Interestingly though, Goldman et al. [13] emphasize that the net change is an overall decline among the migration population. They argue that the potential detriments to health upon migration (e.g., stress, stigmatization) outdo the potential benefits to health upon migration (e.g., greater socioeconomic attainment). Alternatively, the differences in health status years later could be due to migrants having a different reference group (e.g., U.S. inhabitants) than non-migrants (e.g. other non-migrant Mexicans). Determining one's place within a social hierarchy appears to be at play in explaining the results of Goldman et al. [13].

Given this mixed evidence, it is also of great interest to study health selection in a context where the barriers to migration are higher to identify what additional factors are migrants and non-migrants "selecting" on before migration occurs. Furthermore, it is important to determine if health selection persists over time, accounting for factors that could explain selection.

The Philippines provides a particularly informative context in which to study migration and health selection. Compared to other immigrants, Filipinos comprise one of the four visa "oversubscribed" countries to the U.S. which include China, India, and Mexico [31–33]. The number of prospective immigrants requesting visas from these countries outnumber the total number of visas allotted. Immigrants from these countries also make up the bulk of immigration to the U.S. and often face long wait times to arrive as lawful permanent residents [6, 32, 33]. Although immigrants from Mexico continue to comprise the majority of immigrants in the U.S. at 25% of the entire immigrant population (11.2 million), immigrants from Asia represent the fastest growing group of immigrants to the U.S. [34]. Immigrants from the Philippines make up 4% of all immigrants in the U.S. or about 2 million people [35]. As of 2019, over 4.2 million Filipinos live in the U.S., only exceeded by Chinese (5.4 million) and Asian Indian (4.6 million) ethnic groups as the most populous Asian groups [35].

Despite comprising a smaller majority of the overall immigrant population in the U.S., Filipino immigrants occupy a unique space in U.S. immigration history and policy as a former colonial holding of Spain and later the U.S. The Philippines was annexed by the U.S. from Spain after the Spanish-American War and the Philippine-American War. Filipinos in the late 1800s and early 1900s were treated as "U.S. nationals" and allowed to migrate to and from the U.S. without the same restrictions as other Asian ethnic groups [36, 37]. Most Filipinos migrating to the U.S. worked in agriculture and fishing industries in Hawaii, Alaska, and U.S. West Coast. The Philippines also saw intensive development of its healthcare industry, with its schools modeling U.S. medical and nursing schools [37, 38]. When the Philippines was formally granted independence from the U.S. after World War II, Filipinos continued to migrate to the U.S. Policies such as the Immigration and Naturalization Act of 1965 relaxed country quotas for immigration from non-European countries. As a result, many Filipinos immigrated to the U.S., with many of them working in nursing, healthcare, hospitality-related fields [38–40]. Today, the U.S. remains the most popular destination country for Filipino immigrants with over 436,000 Filipinos immigrating to the U.S. between 2007 and 2017, nearly 70% more than the second most popular destination, Canada (over 260,000 Filipino immigrants between 2007 and 2017) [39]. In fact, nearly 77% of all emigration from the Philippines between 2007 and 2017 was to the North and South America or associated territories [39]. Today, many Filipinos who immigrate to the U.S. also arrive via "family reunification" or "employment" visas [6, 37, 39] which are representative the ongoing historical stream of Filipino migration to the U.S. and continued labor recruitment. About 40,000 Filipinos are authorized to come to the U.S. per year [39].

Moreover, as result of this historical imperial and colonial relationship with the U.S., Filipinos tend to be more English proficient, with lower rates of unemployment and poverty compared to other Asian ethnic groups [35, 37, 40]. Filipino migrants specifically tend to be of working age (18–65 years old), college educated, and mostly female [39]. Pre-migration studies on Filipino emigrants have noted how Filipinos with better English proficiency and better socioeconomic attainment tend to be healthier compared Filipino non-migrants with respect to obesity [23] and the number of health conditions [6]. These social determinants of health should prevent worse health among Filipinos compared to other Asian ethnic groups and immigrants over time. However, this is not the case. Studies have noted how Filipinos report worse self-rated health [41, 42], and had greater obesity, high blood pressure, and diabetes compared to other Asian groups and non-Hispanic White people [42, 43]. Thus, Filipinos may not appear as a "healthy" immigrant group despite possessing many of the social determinants of health that should preclude poor health. Some have suggested that the poor health among Filipinos could be related to intergenerational trauma and neocolonial structures that place stress among Filipino immigrants and their economic success in the U.S. [37].

Overall, studying Filipino migration to the U.S. provides the field with an interesting look at how the stresses and successes of migration can affect a more socioeconomic privileged, historically established, and growing immigrant diaspora. This differs in contrast to the overwhelming perception of immigrants as socioeconomically disadvantaged upon arrival to the U.S.

## Methodological limitations of studying migration and health selection

Methodologically, thoroughly studying migration and health selection has been limited for two key reasons. Although it is well acknowledged in the literature that health selection occurs for immigrants, few formal comparisons between migrants and non-migrants exist [6, 12, 13, 22, 23, 44, 45]. The literature on health selection among immigrants has typically been predicated on studies comparing immigrants to those who are native-born within the destination country [19, 20]. There are limitations to this approach because the very idea of selection requires that we consider whether people in a given country who migrate differ than those who remain behind [44, 46]. Comparisons between immigrants and native-born individuals instead provides an analysis of the *effect of nativity*, *rather than the effect of migration*. Thus, conclusions about health selection between immigrant and native-born people are potentially erroneous, because both groups are inherently different by virtue of origin country. Instead, non-migrants in the country of origin serve as a more appropriate comparison group than a group in the destination country [44, 45]. Non-migrants from the country of origin may have had more similar experiences growing up to migrants than those who were born in the U.S. Moreover, comparisons between migrants and non-migrants can better examine the effect of immigrant integration or acculturation [46].

Second, few longitudinal studies on immigrant health exist within the U.S. context. Those that exist have examined the experiences of people from Mexico (e.g., the Mexican Family Life Survey/MxFLS) [22] or immigrants after arrival (e.g., New Immigrant Survey) [47]. Of these studies, only one, the MxFLS, had exclusively measured both pre-migration and post-migration experiences.

This study offers two key strengths in examining migrant health selection. First, we compare migrants to those who did not migrate in the country of origin (non-migrants). A second contribution of this study is that we examine health selection at two time points: pre-migration and 1-year post-migration. Most studies have examined health selection retrospectively [3, 22, 47, 48]. This study design allows us to examine how migrants change compared to non-migrants over time. To do this, we use the Health of Philippine Emigrants Study (HoPES), a

longitudinal cohort study that began in 2017 that compares Philippine emigrants to the U.S. with those who remain in the Philippines [44].

We consider the following hypotheses. First, that migrants will have better SRH compared to non-migrants prior to migration and 1-year follow up. And, second, that differences in SRH between migrants and non-migrants will be largely accounted for by health indicators (e.g., physical and mental health), rather than demographic, socioeconomic, and social factors.

## Materials and methods

### Data

Data are from the Health of Philippine Emigrants Study (HoPES). HoPES is a longitudinal cohort study examining the effects of migration on health. Two cohorts were examined in the parent study. The first consists of migrants emigrating from the Philippines to the U.S. The second cohort consists of non-migrants who intend to remain in the Philippines. Began in 2017, HoPES collected data at five time points: baseline (or pre-migration), 3-month (among the migrant cohort only), 1-year, 2-year, and 3-year. At the time of this study, data were only available for baseline, 3-month, and 1-year data only. We used data from the baseline and 1-year data only as these two follow-up points contained data for both migrant and non-migrant cohorts.

Migrants were recruited at Manila and Cebu offices of the Commission of Filipinos Overseas (CFO), the national emigration regulatory agency of the Philippines government [44]. In order to legally migrate from the Philippines, prospective emigrants must attend a two-hour Pre-Departure Orientation Session (PDOS) at an approved CFO office. Manila and Cebu serve as the two main hubs for all migrants who intend to emigrate to the U.S. or otherwise. The migrant cohort was recruited at PDOS sessions. Migrants 20 to 59 years old at the time of recruitment who intended to migrate to the U.S. in the next three months were eligible to participate. Those who did not speak English, Tagalog, or Cebuano, were not migrating to the U.S. as their final destination, and being pregnant at the time of recruitment were excluded. Pregnant people were excluded given that HoPES was originally a study on obesity.

After recruiting the migrant sample, non-migrants were sampled via stratified random sampling of households in Metro Manila, Metro Cebu-urban, and Metro Cebu-rural and were frequency-matched to migrant participants on age, sex, education, and urbanicity to allow for better comparability [4, 44, 45]. Within each stratification, barangays (e.g., the Philippine equivalent to the U.S. census tract) were randomly chosen to be sampled based on population size. Eight barangays were chosen from Metro Manila. Seven barangays were chosen from Metro Cebu-urban. Finally, five barangays were chosen from Metro Cebu-rural. Within each barangay, households for sampling were chosen by first identifying a random landmark. After the landmark was chosen, interviewers proceeded in a random direction away from the landmark randomly choosing houses to sample. After the number of adult residents were screened for eligibility and enumerated within each household, the interview was then conducted with one chosen member of the household [44, 45]. Non-migrant eligibility criteria included those who had resided in their home for the past two years, did not intend to move from their residence within three years of study recruitment, were currently between the ages of 20 to 59, able to speak Tagalog, Cebuano, or English, were not live-in domestic workers, and were not pregnant. This study was approved by the Institutional Review Boards at the University of California Los Angeles, the USC-Office of Population Studies Foundation, Incorporated, Philippines, and the Ethics Review Committee of the University of San Carlos, Philippines. Further information on recruitment, sampling, and data collection procedures are provided elsewhere [44, 45]. Both migrant and non-migrant participants provided informed consent via written consent. Participants who wanted to participate in HoPES provided their signature as informed

written consent after being informed of the study by trained HoPES staff and having an opportunity to ask clarifying questions. Given this sampling strategy of non-migrants in relation to the migrant cohort and this study's focus on immigration, both migrant and non-migrant samples are intended to be representative of recent Filipino immigrants to the U.S. [44].

## Measures

**Dependent variable.** SRH was asked in the following manner: "Compared to people your age, would you say your health is. . .". Participants responded on a on a 5-point scale (0 = poor, 1 = fair, 2 = good, 3 = very good, 4 = excellent) and analyzed as a continuous variable, similar to previous studies [27, 49]. We chose to examine SRH as a continuous variable because it provides easily interpretable coefficients and adjusted R-squared measures. The use of adjusted R-squared allows us to quantify how much variance in SRH may be accounted for by variables in the model. By looking at nested models, we can further evaluate how much the adjusted R-squared increases with inclusion of blocks of variables.

We supplement our analyses of SRH as a continuous variable by conducting sensitivity analyses examining SRH as a binary outcome (poor/fair vs. good/very good/excellent health) and 5-level ordinal outcome binary outcome (poor/fair vs. good/very good/excellent health) and 5-level ordinal outcome [9].

We considered seven domains that may account for differences in SRH between migrants and non-migrants.

**Demographic factors.** Age in years (continuous) and gender (male/female) were self-identified at baseline and included as common demographic factors that contribute to migrant selection [3, 44]. Age in years was calculated as the difference between the date of interview from participants' reported birth date. Language of interview (any English used/no English used) was noted at both baseline and 1-year, which accounts for potential differences in comprehension and translation [50].

**Physical health.** Physical health was measured using measures consistent with allostatic load, henceforth "allostatic load", defined as the physiological wear-and-tear of the body resultant from stress [51]. Allostatic load was calculated by determining whether participants were at clinical high-risk based on the following biomarkers, measured at baseline only: systolic blood pressure, diastolic blood pressure, body mass index, waist circumference, waist-to-hip ratio, total cholesterol, low density lipoprotein, high density lipoprotein, triglycerides, apolipoprotein-B, and C-reactive protein [51]. Using clinical cutoffs is a common technique for calculating allostatic load [52]. For each biomarker, a value of "1" was assigned if the biomarker was at or above the clinical cutoff and "0" otherwise. These values were summed to create a composite score, with higher scores indicating greater allostatic load (see S1 Table).

**Mental health.** We used three measures of mental health, the Cohen Perceived Stress Scale (range = 1–5) [53], the PROMIS Scale for Emotional Distress (range = 1–5) [54], and the PROMIS Cognitive Abilities and Cognitive Concerns Scales short form (range = 0–16) [55], which have been used either among Filipinos or within multicultural populations [56, 57]. Higher scores on these three scales indicate increased general stress, greater distress, and worse cognitive functioning, respectively. Perceived stress and emotional distress were measured at baseline and 1-year, while cognitive functioning was measured at baseline only.

**Health behaviors.** Physical activity was measured using the International Physical Activity Questionnaire [58], which has been previously used in the Philippines context. Respondents reported the number of hours they spent doing physical activity and how vigorously they performed these activities. Physical activity was categorized as low (sedentary/moderately inactive) or high (moderately active/active).

Diet was assessed using questions that asked about the usual frequency of consumption of fresh fruits and fresh vegetables—foods linked to decreased risk of non-communicable diseases such as cancer and cardiovascular disease [59] and consumption of fast food (e.g. hamburgers, fish fillet sandwich, pizza), and soda/sweetened drinks—foods high in sodium, fat and/or sugar known to increase risk of non-communicable diseases [60]. Our four measures of diet—consumption of fast food, soda or soft drinks, fresh vegetables, and fresh fruit—were measured using a Food Frequency Questionnaire [61], which had been previously used in a pilot study for HoPES [62] and was modified with the input from community partners [44]. These four items were chosen given their salability with healthy eating and as behavioral predictors of obesity, a commonly used indicator of overall health [23]. Items were originally coded as "Never", "Less than once a week", "1–2 times per week", "3–5 times per week", and "6 or more times per week". For simplicity of interpretation, we dichotomized consumption as "less than once a week" and "once a week or more." We coded variables as 1 to indicate healthier dietary behavior (e.g., less than once a week fast food consumption, once a week or more consumption of fresh vegetables) and 0 to indicate unhealthy dietary behavior (e.g., once a week or more fast food consumption, less than once a week fresh vegetable consumption).

We examined two measures of sleep and sleep quality: self-reported hours of sleep (less than 7 hours, 7 to 9 hours, and more than 9 hours) and the PROMIS sleep quality scale (range: 1 = poor to 4 = very good) [63], which has been previously used in multicultural populations [57].

**Socioeconomic factors and healthcare utilization.** Socioeconomic factors included educational attainment (less than high school, high school graduate, some college, or college degree or more) and financial strain (high = some to considerable difficulty meeting expenses, medium = just enough to pay expenses with no difficulty, or low = enough money with some left over) [64]. Financial strain was measured at both baseline and 1-year, while educational attainment as measured only at baseline. We also examine the role of healthcare utilization. Healthcare utilization was determined by asking where the respondent sought medical care/advice when the respondent was last sick or injured (no treatment, hospital, or clinic/other). Healthcare utilization was only measured at baseline.

**Social capital.** Social capital was measured using an adapted Resource Generator scale [65], which has been previously validated in the Philippine context [66]. Participants were asked whether they knew someone that could support them with specific tasks, such as loaning enough money to pay rent/mortgage for one month. Participants were then asked how easy it would be to utilize this resource. We constructed a composite score with higher scores indicating greater social capital (range: 0–12).

Social isolation was determined using the item: "In the past 7 days, has the respondent felt isolated from others?" (low = never/rarely/sometimes or high = often/always). Social capital and isolation were measured at baseline and 1-year.

**Social desirability.** We measured social desirability at baseline using the following: "In the past 7 days, have you said untrue things to avoid being embarrassed?" (low = never/rarely/sometimes or high = often/always).

## Data analysis

Analyses were conducted using Stata 15.0. We restricted our pre-migration analysis to those with a complete set of variables at baseline (n = 1,632 or 99% of the sample). For 1-year follow-up analyses, we restricted the sample to participants with complete data at both baseline and 1-year follow up (n = 1,195 or 73% of the sample).

Post-stratification weights were applied to align the migrant sample with the distribution of age, sex, and education among recent Filipino migrants in the U.S. based on the 2010

American Community Survey [44, 45]. To address potential response bias due to attrition, we used response propensity weight adjustment to reweight the sample to align with baseline characteristics [67]. The response propensities were modeled using logistic regression, with the covariates identified as significant in S2 Table and the inverse of the estimated propensities were used as weights. Both post-stratification weights and propensity weights were applied to the dataset using the "svyset" command in Stata [68, 69].

We first examined differences in each domain by migrant status using t-tests and chi-square tests at both baseline and 1-year. We then fit a series of ordinary least squares (OLS) regression models adding each domain in the following order: migrant status, demographics, physical health, mental health, health behaviors, socioeconomic status and healthcare utilization, psychosocial factors, and social desirability using the "svy" command. The "svy" command produces robust standard errors that account for the complexity of the study design. We repeated analyses for SRH for 1-year follow up substituting any variables that were measured at baseline for the same variable measured at 1-year (e.g., health behaviors). Variables measured only at baseline were unchanged (e.g., allostatic load). We then examined changes in the adjusted R-squared between models to examine how much each domain added to explaining differences in SRH.

In our sensitivity analyses of SRH as a 5-level ordinal outcome, we evaluated whether the proportional odds assumption was met using the "omodel" user written command in Stata [70]. We found that the proportional odds assumption was not violated for our key variable of interest, migrant status, in a crude bivariate model (Likelihood-Ratio Test of Proportionality of Odds: $X^2$ (3 df) = 4.65, p = .200). However, a global test indicated that the proportional odds assumption was violated for one or more variables in the fully adjusted model (Likelihood-Ratio Test of Proportionality of Odds: $X^2$ (58 df) = 92.16, p = .003). Further exploration using the "gologit2" user written command in Stata [71] revealed that migrant status was not one of the variables that violated the proportional odds assumption in the fully adjusted model (p = .462). For this reason and for parsimony, we chose to retain the ordinal logistic regression, instead of exploring other generalized models for our sensitivity analyses [70–72].

## Results

### Descriptive statistics

Table 1 presents the descriptive statistics comparing migrants and non-migrants at baseline and 1-year follow up. At baseline, 832 migrants and 800 non-migrants were surveyed (n = 1632 total) [45]. At the 1-year follow up post-migration (for migrants), 52.9% of migrants (n = 440) and 94.3% of non-migrants (n = 754) were retained. Compared to migrants who were retained, migrants missing at 1-year had better SRH and were more educated. Migrants missing at 1-year were older and had slightly greater financial strain compared to migrants who remained (S2 Table). In the analytical sample, migrants had better SRH (Mean = 2.15) compared to non-migrants at baseline (Mean = 1.38); differences remained at 1-year.

There was little difference in baseline allostatic load by migrant status. However, migrants reported less emotional distress and perceived stress, and better cognitive functioning compared to non-migrants at both baseline and 1-year follow-up.

Migrants consumed fewer soft drinks and more fruits than non-migrants at both baseline and 1-year. More migrants slept between 7-to-9 hours per night and had better sleep quality than non-migrants at both baseline and 1-year. Though similar at baseline, migrants consumed more vegetables at 1-year than non-migrants.

Migrants had greater educational attainment at baseline compared to non-migrants at baseline and 1-year follow-up. More migrants had low baseline financial strain compared to non-

**Table 1. Summary statistics for HoPES sample at baseline and 1-year follow-up.**

| Variable (range) | Panel A: Baseline Sample (n = 1632) | | | Panel B: Year-1 Sample (n = 1194) | | |
|---|---|---|---|---|---|---|
| | Migrant (N = 832) % or mean (SE) | Non-migrant (N = 800) % or mean (SE) | p-value | Migrant (N = 440) % or mean (SE) | Non-migrant (N = 754) % or mean (SE) | p-value |
| Self-rated health (0–4) | 2.15 (0.03) | 1.38 (0.03) | < .001 | 2.53 (0.06) | 1.34 (0.03) | < .001 |
| Good/Very Good/Excellent Health | 73.0% | 35.4% | < .001 | 76.6% | 32.4% | < .001 |
| **Demographic Domain** | | | | | | |
| Age | 37.03 (0.41) | 36.89 (0.41) | .807 | 35.00 (0.52) | 37.66 (0.42) | < .001 |
| Male[1] | 33.6% | 33.6% | .988 | 32.3% | 33.5% | .678 |
| Interview language, any English | 12.9% | 4.5% | < .001 | 69.7% | 2.4% | < .001 |
| **Physical Health Domain** | | | | | | |
| Allostatic load (0–9)[1] | 2.64 (0.06) | 2.74 (0.07) | .303 | 2.44 (0.09) | 2.81 (0.07) | .001 |
| **Mental Health Domain** | | | | | | |
| Emotional distress (1–5) | 1.32 (0.02) | 1.73 (0.03) | < .001 | 1.34 (0.03) | 1.67 (0.03) | < .001 |
| Perceived stress (0–5) | 1.59 (0.02) | 2.16 (0.02) | < .001 | 1.51 (0.03) | 2.12 (0.02) | < .001 |
| Cognitive functioning (0–16)[1,2] | 2.62 (0.10) | 4.75 (0.11) | < .001 | 2.52 (0.14) | 4.66 (0.11) | < .001 |
| **Health Behavior Domain** | | | | | | |
| High physical activity | 82.6% | 95.9% | < .001 | 84.4% | 80.9% | .129 |
| Low fast food consumption | 51.5% | 50.4% | .664 | 40.8% | 46.5% | .058 |
| Low soft drink consumption | 48.0% | 26.1% | < .001 | 40.0% | 31.9% | .005 |
| High fresh vegetable consumption | 91.4% | 91.5% | .922 | 91.6% | 85.8% | .004 |
| High fresh fruit consumption | 90.8% | 87.6% | .040 | 94.7% | 85.5% | < .001 |
| Hours of sleep per night | | | < .001 | | | < .001 |
| Less than 7 hours | 18.4% | 28.1% | | 30.0% | 26.7% | |
| 7-to-9 hours | 64.1% | 57.3% | | 62.8% | 57.9% | |
| More than 9 hours | 17.5% | 14.6% | | 7.2% | 15.5% | |
| Sleep quality | 2.71 (0.03) | 2.08 (0.03) | < .001 | 2.87 (0.05) | 2.10 (0.03) | < .001 |
| **Socioeconomic and Healthcare Utilization Domain** | | | | | | |
| Educational attainment[2] | | | < .001 | | | < .001 |
| Less than high school | 8.0% | 12.8% | | 4.6% | 15.8% | |
| High school graduate | 20.9% | 16.2% | | 18.3% | 19.4% | |
| Some college | 18.2% | 37.3% | | 15.6% | 33.9% | |
| College degree or more | 52.9% | 33.7% | | 61.5% | 30.9% | |
| Financial strain [3] | | | < .001 | | | < .001 |
| High | 17.5% | 42.9% | | 6.7% | 37.8% | |
| Medium | 55.6% | 46.8% | | 29.3% | 49.1% | |
| Low | 26.9% | 10.3% | | 64.0% | 13.1% | |
| Healthcare Utilization[1] | | | < .001 | | | < .001 |
| No treatment | 49.8% | 34.0% | | 50.9% | 35.0% | |
| Hospital | 32.0% | 34.4% | | 30.2% | 32.8% | |
| Clinic or other | 18.3% | 31.6% | | 18.9% | 32.3% | |
| **Social Capital** | | | | | | |
| Social capital (0–12) | 8.01 (0.08) | 7.29 (0.09) | < .001 | 7.89 (0.13) | 8.54 (0.10) | < .001 |
| High social isolation | 9.2% | 20.7% | < .001 | 14.8% | 16.9% | .332 |
| **Social Desirability** | | | | | | |
| High social desirability[1,4] | 10.3% | 20.6% | < .001 | 9.9% | 19.7% | < .001 |

[1] Variable measured at baseline only.

[2] Higher score on the "cognitive functioning" scale indicates poorer cognitive functioning.

[3] High financial strain indicates that participants had "Some to considerable difficulty in meeting expenses". Medium financial strain indicates that participants had "just enough to pay expenses without difficulty". Low financial strain indicates that participants had "enough money with money leftover".

[4] High social desirability refers to people who "sometimes", "often", or "always" said untrue things to avoid being embarrassed.

migrants with a similar advantage at 1-year follow-up. More migrants reported not seeking treatment when they were last ill compared to non-migrants. These baseline differences in healthcare utilization were present among those included at 1-year follow-up.

Migrants reported greater baseline social capital compared to non-migrants. However, migrants reported lower social capital at 1-year follow-up compared to non-migrants. Migrants reported lower social isolation at baseline compared to non-migrants, though there were few differences between those who remained at 1-year.

## Regression of baseline self-rated health on migrant status and health domains

Table 2 displays the multivariable OLS regression of migrant status and other domains on baseline SRH. With no controls (Model 1), migrants reported better SRH compared to non-migrants ($\beta$ = 0.77, 95% CI = 0.68, 0.86). Migrant status alone accounted for 15.1% of the total variance in SRH. Accounting for demographic factors (Model 2), migrants reported better SRH compared to non-migrants ($\beta$ = 0.74, 95% CI = 0.65, 0.83). Migrant status and demographic factors accounted for 18.2% of the variance in SRH.

While allostatic load (Model 3) provided little change in the adjusted R-squared of SRH, the inclusion of mental health (Model 4) increased the adjusted R-squared to 24.8%. Migrants still reported better SRH compared to non-migrants, albeit at a reduced magnitude ($\beta$ = 0.50, 95% CI = 0.40, 0.60). Increased allostatic load, emotional distress, perceived stress, and poorer cognitive were associated with poorer SRH.

Including health behavior (Model 5) reduced the magnitude of the migrant advantage ($\beta$ = 0.35, 95% CI = 0.24, 0.44) and increased the adjusted R-squared to 33.3%. While all of the indicators of physical and mental health remained associated with poorer SRH, greater physical activity was also associated with poorer SRH. Fast food, soda, fruit, and vegetable consumption were not highly associated with SRH. Compared to sleeping 7-to-9 hours per night, sleeping less than 7-hours per night was associated with better health while sleeping for more than 9-hours per night was associated with poorer SRH. Higher sleep quality was associated with better sleep.

Migrants still reported better health compared to non-migrants after including socioeconomic factors and healthcare utilization (Model 6) ($\beta$ = 0.30, 95% CI = 0.20, 0.40). The adjusted R-squared slightly increased to 35.0%. Having a college degree and above (relative to less than a high school education) and low financial strain (relative to high financial strain) were associated with better SRH. Utilizing a hospital the last time a participant was sick or injured was associated with poorer SRH compared to seeking no treatment at all.

Although the magnitude of the migrant advantage increased with the inclusion of social capital (Model 7) ($\beta$ = 0.32, 95% CI = 0.22, 0.43), there was little change in the adjusted R-squared (35.1%). Finally, the differences between migrants and non-migrants remained the same when including social desirability (Model 8) ($\beta$ = 0.32, 95% CI = 0.22, 0.43) with little change in adjusted R-squared (35.2%). Higher social desirability was associated with poorer SRH.

## Regression of 1-year self-rated health on migrant status and health domains

Table 3 presents the multivariable OLS regression results of 1-year SRH on migrant status and the health domains at 1-year (except factors that were only measured at baseline). The 1-year data follow similar trends as the baseline analysis. Migrant status alone accounted for 26.2% of the total variance in SRH. Migrants at 1-year reported better SRH compared to non-migrants

**Table 2. Multivariable ordinary least squares regression of baseline self-rated health on migrant status, Health of Philippine Emigrant Study (HoPES), N = 1632.**

| Variables | Model 1 β | Model 1 95% CI | Model 2 β | Model 2 95% CI | Model 3 β | Model 3 95% CI | Model 4 β | Model 4 95% CI | Model 5 β | Model 5 95% CI | Model 6 β | Model 6 95% CI | Model 7 β | Model 7 95% CI | Model 8 β | Model 8 95% CI |
|---|---|---|---|---|---|---|---|---|---|---|---|---|---|---|---|---|
| **Migrant Status (Non-migrant ref.)** | 0.77 | 0.68, 0.86 | 0.74 | 0.65, 0.83 | 0.74 | 0.65, 0.83 | 0.50 | 0.40, 0.60 | 0.35 | 0.25, 0.44 | 0.30 | 0.20, 0.40 | 0.32 | 0.22, 0.43 | 0.32 | 0.22, 0.43 |
| **Demographic Domain** | | | | | | | | | | | | | | | | |
| Age | | | -0.01 | -0.02, -0.01 | -0.01 | -0.01, -0.01 | -0.01 | -0.02, -0.01 | -0.01 | -0.01, -0.01 | -0.01 | -0.01, -0.00 | -0.01 | -0.01, -0.00 | -0.01 | -0.01, -0.00 |
| Gender (Female ref.) | | | | | | | | | | | | | | | | |
| Male | | | 0.03 | -0.06, 0.13 | 0.01 | -0.09, 0.10 | -0.03 | -0.12, 0.06 | -0.01 | -0.10, 0.08 | 0.01 | -0.08, 0.09 | 0.01 | -0.07, 0.10 | 0.02 | -0.07, 0.10 |
| Interview Language (No English ref.) | | | | | | | | | | | | | | | | |
| Any English | | | 0.32 | 0.15, 0.49 | 0.33 | 0.16, 0.50 | 0.32 | 0.16, 0.48 | 0.25 | 0.09, 0.41 | 0.21 | 0.05, 0.37 | 0.21 | 0.05, 0.37 | 0.21 | 0.05, 0.37 |
| **Physical Health** | | | | | | | | | | | | | | | | |
| Allostatic load | | | | | -0.05 | -0.07, -0.02 | -0.05 | -0.08, -0.03 | -0.05 | -0.07, -0.03 | -0.05 | -0.07, -0.02 | -0.05 | -0.07, -0.02 | -0.05 | -0.07, -0.02 |
| **Mental Health** | | | | | | | | | | | | | | | | |
| Emotional distress | | | | | | | -0.10 | -0.18, -0.03 | -0.04 | -0.11, 0.03 | -0.02 | -0.10, 0.05 | -0.01 | -0.09, 0.06 | -0.01 | -0.08, 0.07 |
| Perceived stress | | | | | | | -0.15 | -0.23, -0.07 | -0.12 | -0.20, -0.04 | -0.10 | -0.18, -0.02 | -0.10 | -0.18, -0.02 | -0.09 | -0.17, -0.02 |
| Cognitive functioning[1] | | | | | | | -0.05 | -0.07, -0.04 | -0.04 | -0.05, -0.02 | -0.03 | -0.05, -0.02 | -0.03 | -0.05, -0.02 | -0.03 | -0.05, -0.02 |
| **Health Behavior Domain** | | | | | | | | | | | | | | | | |
| Physical Activity (Low ref.) | | | | | | | | | | | | | | | | |
| High physical activity | | | | | | | | | -0.17 | -0.30, -0.04 | -0.17 | -0.30, -0.04 | -0.18 | -0.31, -0.05 | -0.18 | -0.31, -0.05 |
| Fast Food consumption (More than once a week ref.) | | | | | | | | | | | | | | | | |
| Less than once a week | | | | | | | | | -0.09 | -0.17, -0.00 | -0.07 | -0.16, 0.02 | -0.07 | -0.15, 0.02 | -0.07 | -0.15, 0.02 |
| Soda Consumption (More than once a week ref.) | | | | | | | | | | | | | | | | |
| Less than once a week | | | | | | | | | 0.02 | -0.07, 0.11 | 0.01 | -0.08, 0.10 | 0.01 | -0.08, 0.10 | 0.01 | -0.08, 0.10 |
| Vegetable Consumption (Less than once a week ref.) | | | | | | | | | | | | | | | | |
| More than once a week | | | | | | | | | 0.07 | -0.07, 0.21 | 0.07 | -0.07, 0.20 | 0.06 | -0.08. 0.19 | 0.06 | -0.08, 0.19 |
| Fruit Consumption (Less than once a week ref.) | | | | | | | | | | | | | | | | |
| More than once a week | | | | | | | | | 0.09 | -0.04, 0.23 | 0.09 | -0.04, 0.22 | 0.09 | -0.05, 0.22 | 0.09 | -0.05, 0.22 |
| Hours of Sleep (7-to-9 hours ref.) | | | | | | | | | | | | | | | | |
| Less than 7 hours | | | | | | | | | 0.19 | 0.09, 0.29 | 0.17 | 0.07, 0.27 | 0.17 | 0.07, 0.27 | 0.17 | 0.07, 0.28 |
| More than 9 hours | | | | | | | | | -0.13 | -0.24, -0.01 | -0.12 | -0.23, -0.00 | -0.12 | -0.24, -0.00 | -0.12 | -0.23, -0.00 |
| Sleep Quality | | | | | | | | | 0.37 | 0.31, 0.43 | 0.36 | 0.30, 0.42 | 0.36 | 0.30, 0.42 | 0.36 | 0.30, 0.42 |
| **Socioeconomic and Healthcare Utilization Domain** | | | | | | | | | | | | | | | | |
| Educational Attainment (less than high school ref.) | | | | | | | | | | | | | | | | |
| High school graduate | | | | | | | | | | | -0.01 | -0.15, 0.14 | -0.01 | -0.16, 0.14 | -0.01 | -0.15, 0.14 |
| Some college | | | | | | | | | | | 0.10 | -0.04, 0.25 | 0.09 | -0.06, 0.24 | 0.09 | -0.05, 0.24 |

(*Continued*)

**Table 2.** (Continued)

| Variables | Model 1 β | Model 1 95% CI | Model 2 β | Model 2 95% CI | Model 3 β | Model 3 95% CI | Model 4 β | Model 4 95% CI | Model 5 β | Model 5 95% CI | Model 6 β | Model 6 95% CI | Model 7 β | Model 7 95% CI | Model 8 β | Model 8 95% CI |
|---|---|---|---|---|---|---|---|---|---|---|---|---|---|---|---|---|
| College degree and above | | | | | | | | | | | 0.16 | 0.02, 0.30 | 0.15 | 0.01, 0.29 | 0.15 | 0.01, 0.29 |
| **Financial Strain (High ref.)[1]** | | | | | | | | | | | | | | | | |
| Medium | | | | | | | | | | | 0.09 | -0.01, 0.18 | 0.08 | -0.01, 0.19 | 0.08 | -0.01, 0.18 |
| Low | | | | | | | | | | | 0.31 | 0.18, 0.44 | 0.30 | 0.17, 0.43 | 0.30 | 0.17, 0.43 |
| **Healthcare Utilization (No treatment ref.)** | | | | | | | | | | | | | | | | |
| Hospital | | | | | | | | | | | -0.10 | -0.19, -0.01 | -0.10 | -0.20, -0.01 | -0.11 | -0.20, -0.02 |
| Clinic or other | | | | | | | | | | | -0.07 | -0.18, 0.03 | -0.08 | -0.18, 0.03 | -0.08 | -0.18, 0.03 |
| **Social Capital** | | | | | | | | | | | | | | | | |
| Social Capital | | | | | | | | | | | 0.01 | -0.01, 0.04 | 0.01 | -0.01, 0.04 | | |
| **Social isolation (Low ref.)** | | | | | | | | | | | | | | | | |
| High social isolation | | | | | | | | | | | -0.13 | -0.34, 0.08 | -0.10 | -0.31, 0.10 | | |
| **Social Desirability (Low ref.)[3]** | | | | | | | | | | | | | | | | |
| High social desirability | | | | | | | | | | | | | | | -0.28 | -0.52, -0.05 |
| Constant | 1.38 | 1.32, 1.44 | 1.81 | 1.66, 1.97 | 1.83 | 1.67, 1.98 | 2.74 | 2.47, 3.01 | 1.68 | 1.30, 2.05 | 1.40 | 1.00, 1.80 | 1.25 | 0.80, 1.70 | 1.23 | 0.78, 1.68 |
| R-squared | 0.151 | | 0.182 | | 0.188 | | 0.248 | | 0.333 | | 0.350 | | 0.351 | | 0.352 | |

Note. Analysis were done with non-response weighting to accounting for missingness.

[1] Higher cognitive functioning score indicates poorer cognitive functioning.

[2] High financial strain indicates that participants had "Some to considerable difficulty in meeting expenses". Medium financial strain indicates that participants had "just enough to pay expenses without difficulty". Low financial strain indicates that participants had "enough money with money leftover".

[3] High social desirability refers to people who "sometimes", "often", or "always" said untrue things to avoid being embarrassed.

remaining in the sample, even when accounting for each domain (Model 8: β = 0.28, 95% CI = 0.10, 0.46). The adjusted R-squared of the final model (47.6%) was higher compared to the same model using baseline data (34.0%). The inclusion of health behaviors saw the largest change in adjusted R-squared, from 33.4% to 44.4%.

## Sensitivity analyses

S3–S6 Tables show the results operationalizing SRH as a binary and as an ordinal outcome at baseline (S3 and S4 Tables) and 1-year (S5 and S6 Tables). The results were similar to the results obtained by OLS regression.

## Discussion

Our study examined the healthy migrant effect using baseline and 1-year follow up data from HoPES. Previous studies have been limited in that few provided a comprehensive look at factors that could explain differences in health between migrants and non-migrants (e.g., physical health, mental health, health behaviors and social factors). Second, previous studies on health selection relied on cross-sectional analyses or compared migrants with native-born inhabitants

**Table 3. Multivariable ordinary least squares regression of 1-year self-rated health on migrant status, Health of Philippine Emigrants Study (HoPES), N = 1194.**

| Variables | Model 1 | | Model 2 | | Model 3 | | Model 4 | | Model 5 | | Model 6 | | Model 7 | | Model 8 | |
|---|---|---|---|---|---|---|---|---|---|---|---|---|---|---|---|---|
| | β | 95% CI | β | 95% CI | β | 95% CI | β | 95% CI | β | 95% CI | β | 95% CI | β | 95% CI | β | 95% CI |
| **Migrant Status (Non-migrant ref.)** | 1.18 | 1.06, 1.31 | 0.78 | 0.60, 0.96 | 0.77 | 0.59, 0.96 | 0.47 | 0.28, 0.67 | 0.35 | 0.18, 0.53 | 0.24 | 0.06, 0.42 | 0.28 | 0.10, 0.46 | 0.28 | 0.10, 0.46 |
| **Demographic Domain** | | | | | | | | | | | | | | | | |
| Age | | | -0.01 | -0.02, -0.00 | -0.01 | -0.01, -0.00 | -0.01 | -0.01, -0.00 | -0.01 | -0.01, -0.00 | -0.01 | -0.01, -0.00 | -0.01 | -0.01, -0.00 | -0.01 | -0.01, -0.00 |
| Gender (Female ref.)[1] | | | | | | | | | | | | | | | | |
| Male | | | 0.06 | -0.06 0.18 | 0.03 | -0.09, 0.16 | -0.02 | -0.15, 0.10 | -0.04 | -0.15, 0.07 | -0.03 | -0.14, 0.07 | -0.03 | -0.13, 0.08 | -0.02 | -0.13, 0.08 |
| Interview Language (No English ref.) | | | | | | | | | | | | | | | | |
| Any English | | | 0.57 | 0.36, 0.78 | 0.56 | 0.35, 0.78 | 0.54 | 0.34, 0.75 | 0.36 | 0.17, 0.54 | 0.37 | 0.19, 0.54 | 0.37 | 0.19, 0.55 | 0.37 | 0.19, 0.55 |
| **Physical Health** | | | | | | | | | | | | | | | | |
| Allostatic load[1] | | | | | -0.04 | -0.08, -0.01 | -0.05 | -0.08, -0.02 | -0.04 | -0.07, -0.01 | -0.04 | -0.07, -0.01 | -0.04 | -0.07, -0.01 | -0.04 | -0.07, -0.01 |
| **Mental Health** | | | | | | | | | | | | | | | | |
| Emotional distress | | | | | | | -0.14 | -0.22, -0.06 | -0.07 | -0.15, 0.00 | -0.07 | -0.14, 0.01 | -0.06 | -0.14, 0.02 | -0.06 | -0.14, 0.024 |
| Perceived stress | | | | | | | -0.33 | -0.43, -0.23 | -0.22 | -0.32, -0.12 | -0.16 | -0.26, -0.06 | -0.15 | -0.25, -0.05 | -0.15 | -0.25, -0.05 |
| Cognitive functioning[1,2] | | | | | | | -0.03 | -0.05, -0.01 | -0.02 | -0.04, -0.00 | -0.02 | -0.04, -0.00 | -0.02 | -0.04, -0.00 | -0.02 | -0.04, -0.00 |
| **Health Behavior Domain** | | | | | | | | | | | | | | | | |
| Physical Activity (Low ref.) | | | | | | | | | | | | | | | | |
| High physical activity | | | | | | | | | 0.06 | -0.06, 0.18 | 0.07 | -0.05, 0.19 | 0.07 | -0.05, 0.19 | 0.07 | -0.06, 0.19 |
| Fast Food consumption (More than once a week ref.) | | | | | | | | | | | | | | | | |
| Less than once a week | | | | | | | | | -0.04 | -0.14, 0.06 | -0.00 | -0.10, 0.10 | -0.00 | -0.10, 0.10 | -0.00 | -0.10, 0.10 |
| Soda Consumption (More than once a week ref.) | | | | | | | | | | | | | | | | |
| Less than once a week | | | | | | | | | 0.03 | -0.08, 0.14 | 0.04 | -0.07, 0.14 | 0.05 | -0.06, 0.15 | 0.04 | -0.06, 0.15 |
| Vegetable Consumption (Less than once a week ref.) | | | | | | | | | | | | | | | | |
| More than once a week | | | | | | | | | 0.04 | -0.11, 0.19 | 0.07 | -0.08, 0.22 | 0.06 | -0.09, 0.21 | 0.06 | -0.09, 0.21 |
| Fruit Consumption (Less than once a week ref.) | | | | | | | | | | | | | | | | |
| More than once a week | | | | | | | | | -0.09 | -0.24, 0.07 | -0.09 | -0.24, 0.06 | -0.09 | -0.24, 0.06 | -0.09 | -0.24, 0.06 |
| Hours of Sleep (7-to-9 hours ref.) | | | | | | | | | | | | | | | | |
| Less than 7 hours | | | | | | | | | 0.04 | -0.08, 0.15 | 0.02 | -0.09, 0.14 | 0.02 | -0.10, 0.14 | 0.02 | -0.10, 0.14 |
| More than 9 hours | | | | | | | | | -0.09 | -0.24, 0.06 | -0.08 | -0.23, 0.07 | -0.08 | -0.24, 0.07 | -0.08 | -0.23, 0.07 |
| Better Sleep Quality | | | | | | | | | 0.44 | 0.38, 0.51 | 0.42 | 0.36, 0.49 | 0.41 | 0.35, 0.48 | 0.41 | 0.35,— 0.48 |
| **Socioeconomic and Healthcare Utilization Domain** | | | | | | | | | | | | | | | | |
| Educational Attainment (less than high school ref.)[1] | | | | | | | | | | | | | | | | |
| High school graduate | | | | | | | | | | | -0.01 | -0.18, 0.16 | -0.01 | -0.18, 0.16 | -0.01 | -0.17, 0.16 |
| Some college | | | | | | | | | | | 0.09 | -0.07, 0.25 | 0.08 | -0.08, 0.24 | 0.09 | -0.07, 0.25 |

*(Continued)*

**Table 3.** (Continued)

| Variables | Model 1 | | Model 2 | | Model 3 | | Model 4 | | Model 5 | | Model 6 | | Model 7 | | Model 8 | |
|---|---|---|---|---|---|---|---|---|---|---|---|---|---|---|---|---|
| | β | 95% CI | β | 95% CI | β | 95% CI | β | 95% CI | β | 95% CI | β | 95% CI | β | 95% CI | β | 95% CI |
| College degree and above | | | | | | | | | | | -0.02 | -0.18, 0.14 | -0.03 | -0.19, 0.13 | -0.03 | -0.19, 0.13 |
| Financial Strain (High ref.)[3] | | | | | | | | | | | | | | | | |
| Medium | | | | | | | | | | | 0.11 | -0.01, 0.22 | 0.11 | -0.01, 0.22 | 0.11 | -0.01, 0.22 |
| Low | | | | | | | | | | | 0.41 | 0.25, 0.57 | 0.39 | 0.23, 0.55 | 0.39 | 0.23, 0.55 |
| Healthcare Utilization (No treatment ref.)[1] | | | | | | | | | | | | | | | | |
| Hospital | | | | | | | | | | | 0.02 | -0.09, 0.14 | 0.02 | -0.10, 0.13 | 0.01 | -0.10, 0.13 |
| Clinic or other | | | | | | | | | | | -0.02 | -0.14, 0.11 | -0.02 | -0.15, 0.10 | -0.02 | -0.15, 0.10 |
| **Social Capital** | | | | | | | | | | | | | | | | |
| Social Capital | | | | | | | | | | | | | 0.02 | 0.00, 0.04 | 0.02 | 0.00, 0.04 |
| Social isolation (Low ref.) | | | | | | | | | | | | | | | | |
| High social isolation | | | | | | | | | | | | | -0.08 | -0.40, 0.23 | -0.08 | -0.39, 0.23 |
| **Social Desirability (Low ref.)[1,4]** | | | | | | | | | | | | | | | | |
| High social desirability | | | | | | | | | | | | | | | -0.13 | -0.55, 0.29 |
| Constant | 1.34 | 1.28, 1.40 | 1.66 | 1.46, 1.87 | 1.68 | 1.47, 1.88 | 2.86 | 2.56, 3.15 | 1.51 | 1.11, 1.92 | 1.17 | 0.70, 1.64 | 0.99 | 0.51, 1.48 | 0.99 | 0.51, 1.48 |
| R-squared | 0.262 | | 0.298 | | 0.302 | | 0.365 | | 0.459 | | 0.474 | | 0.476 | | 0.476 | |

Note. Analysis were done with non-response weighting to accounting for missingness.

[1] Variable measured at baseline only.

[2] Higher cognitive functioning score indicates worse cognitive functioning.

[3] High financial strain indicates that participants had "Some to considerable difficulty in meeting expenses". Medium financial strain indicates that participants had "just enough to pay expenses without difficulty". Low financial strain indicates that participants had "enough money with money leftover".

[4] High social desirability refers to people who "sometimes", "often", or "always" said untrue things to avoid being embarrassed.

when a more appropriate comparison would be to compare migrants to non-migrants. We addressed limitations encountered by previous studies on the healthy migrant effect by 1) examining if migrants had better SRH compared to non-migrants both before and one year after migration; and 2) identifying the factors underlying differences in SRH by migrant status.

Migrants had a robust advantage for SRH relative to non-migrants at baseline and 1-year follow up. These results corroborate previous studies emphasizing selection on SRH [1, 3]. However, unlike other studies that examined selection post-migration only, our findings show that health selection occurred both prior to and after migration. Thus, health selection is not only a phenomenon after migration. It is also important to consider its processes before migration. For 1-year follow-up specifically, the magnitude of the migrant health advantage was more pronounced, potentially indicating a divergence in health trajectories between both groups. Knowing baseline health for both groups is important in examining if changes in migrant health are due to prevailing theories of acculturation [73] or instead driven by secular trends and globalization [23, 44, 45].

Second, we examined which factors underlay differences in SRH by migrant status at the two time points. Migrants had an advantage over non-migrants in mental health, health behavior, and socioeconomic domains at both baseline and 1-year. However, only mental health and health behaviors contributed to the largest increases in variance explained during multivariable analyses, leaving much of the variance unexplained. Although the remaining domains may also be important in explaining variance in SRH, their contribution may be minimal due to sharing variance with migrant status, mental health, and health behavior [1–3]. Our study also shows how SRH as a global indicator of health for migrants captures not only physical and mental health, but also health behaviors and differences in socioeconomic status. These factors present a more holistic view on health, where one's actions and socioeconomic wellbeing also matter. In the migrant context specifically, the association of better health behaviors and better socioeconomic status may help to explain why the healthy migrant effect may be so persistent. Migrants could have developed healthier behaviors in anticipation for the stresses of the migration process or perhaps to become more similar to their host country counterparts. Gee et al. [23] present this process as "pre-acculturation". In the case of Filipino migrants, their healthier food consumption could be a way in which they acculturate to perceived healthiness of their host country. Pre-acculturation, Gee et al. [23] argue, is bidirectional, as both migrants' and non-migrants' already established social connections in the U.S. could send information on health practices via telephone, internet, and other social media. More work will be needed to see if pre-acculturation indeed occurs longitudinally and if pre-acculturation can grant better health before migration.

While we found that 35.2% of the variance in SRH at baseline and 47.6% was explained by migrant status and the various demographic, social, socioeconomic, healthcare utilization, health status, and health behaviors, there may be additional factors that could account for the unmeasured variance and persistent migrant health advantage. For example, the unexplained variance in SRH could be due to cultural and historical factors among Filipinos that allow them to better evaluate their health. Although SRH has been shown to be a robust measure in multiple communities [74, 75], there may be cultural and historical factors that relate to SRH that cannot be captured through language [50]. For example, the Filipinos may have an intergenerational sense of indebtedness to the economic and material improvements in health and technology among Filipinos brought by U.S. colonialism [37, 76, 77]. This sense of intergenerational indebtedness could manifest as low reports of "fair or poor health" among Filipinos. For migrants especially, movement to the U.S. may symbolically reflect an even greater improvement in life chances, and thus self-rated health.

Another possible factor could be participants' personal affect regarding their health. It is possible that having a positive affect could influence participants' evaluation of their health to be more favorable. Related to this this personal affect, the persisting advantage in SRH could reflect a "honeymoon effect." These migrants are still relatively new to the U.S., and therefore may appraise their health as better because of the successful completion of migration. It is important to consider Goldman et al. [13], who speculate that the apparent improvements to migrant health may also be the result of improved socioeconomic attainment. In our study, we see that migrants are reporting less financial strain compared to non-migrants at both baseline and 1-year follow-up. However, we do not know how long this socioeconomic and health advantage may persist. Other research shows that this advantage may lessen with more time in the U.S. [13, 73]. Will the socioeconomic prospects of migrants continue to improve over time, or will other factors lead to greater financial strain (e.g., remittances)? Future work following migrants over a longer period of time, and considering theoretically relevant factors contributing to the acculturation and integration process would be necessary to determine if this is the case. Moreover, qualitative work should further examine what migrants perceive as

a "successful migration". While the literature has explored "failed" migration in relation to health (i.e. those with worse health are more apt to return to their sending country) [12] or socioeconomic attainment, there remains a dearth in the literature to examine how "successful migration" could be related to health. Finally, it is also important to revisit the question on reference groups when asking about self-rated health. Although the question on self-rated health asked participants to rate their health compared to people of their age, HoPES did not explicitly ask who else participants were comparing themselves to, a limitation also experienced by Goldman et al [13]. Were migrants at 1-year comparing themselves to non-migrants in the Philippines or those who live in the U.S.? Alternatively, are non-migrants at 1-year comparing themselves to other non-migrants or migrants in the U.S.? Although we accounted for possible social desirability, we saw that social desirability was not significantly associated with SRH. Future studies could consider expanding the wording of questions asking about self-rated health to consider more of these relational perspectives in addition to what factors participants account for in evaluating their health.

## Limitations

These findings have limitations. First, the health advantage could be due to attrition in the migrant cohort, potentially biasing the results. Migrants missing at 1-year reported worse baseline SRH compared to those included in the sample (S2 Table). Previous work has shown that people who remain in longitudinal studies tend to be healthier than those who drop out [78]. Nevertheless, we use propensity weighting to upweight those who remained in our two independent cross sectional analyses [67], which allowed us to account for potential contributors to missingness and allowed the 1-year sample to be similarly representative of recent Filipino immigrants to the U.S as baseline [44, 45]. Had attrition not occurred, we would expect that the migrant health advantage would be attenuated. Moreover, we only used two time points to examine a potential health advantage over time by migrant status. While the differences between migrants and non-migrants is stark, it is uncertain if this migrant health advantage will persist in additional follow up waves of HoPES. Future analyses should consider using additional waves of HoPES data as they become available to evaluate if this health advantage remains robust. However, these analyses should consider how further attrition may further bias the health advantage in migrants' favor. Mixed model regressions may serve as an appropriate analysis technique in future longitudinal studies using HoPES data.

Second, we are limited by who is represented in this sample. HoPES recruited only permanent legal migrants to the U.S. and not temporary migrants, asylum seekers, nor undocumented migrants [44, 45]. Thus, these findings are likely generalizable to legal permanent residents, and may not capture the experiences of migrants with other legal statuses. According to the Migration Policy Institute, approximately 313,000 Filipinos were undocumented immigrants in the U.S. or about 3% of all undocumented immigrants (11.3 million) [79]. Previous studies have noted that undocumented immigrants report worse self-rated health compared to other documented immigrants, naturalized citizens, and their U.S.-born counterparts [80]. However, Bacong [41] found that non-citizen Filipinos (which includes both lawful permanent residents, temporary visa holders, and undocumented immigrants) had similar levels of fair/poor self-rated health compared to naturalized and U.S.-born Filipinos [41]. Similar, non-significant findings have been found in nationally representative studies looking at Asian undocumented immigrants in general [81]. However, given the growing literature exploring the importance of legal status on health [2, 6, 41, 80, 81], these structurally vulnerable groups should be considered in future studies.

Finally, while using OLS regression allows for us to have an overall understanding of the contribution of migrant status and associated covariates on differences in SRH, other techniques, such as SEM, should be considered to further tease out the contribution of each factor [82, 83]. SEM can formally test potential mediation pathways and the joint contribution of each factor on SRH. Furthermore, using OLS may have additional statistical issues related to qualitative interpretation of the coefficients. For example, a 1-unit change from "poor" to "fair" health may be qualitatively different than a one-unit change from "very good" to "excellent health". It is also possible that a one-unit change may also be qualitatively different by migrant status. A panel study of recent immigrants to the Netherlands found that more immigrants rated their health as "good" rather than "very good" only a few years post-migration [84]. The authors concluded that although immigrants continued to maintain "good" health upon their migration, unmet expectations of immigration, greater involvement in hazardous work, homesickness, and discrimination from Dutch inhabitants contributed to this downward transition. However, we show in our supplemental analyses (see S3–S6 Tables) that trends between migrants and non-migrants remained the same, even after accounting for possible social and psychological factors. Migrants continued to have a health advantage and report better SRH compared to non-migrants regardless of operationalization of SRH. Ultimately, our use of OLS provides a starting point for future research to examine the effects of these factors and additional health and social factors in future waves of HoPES.

## Conclusion

In conclusion, this article provides further evidence of migrant health selection and adds nuanced understanding to what is being captured by SRH in studies of migrant health. In this case, SRH captures health behaviors and socioeconomic outcomes. These two factors are especially important in considering the social and cultural integration processes that immigrants undergo upon entering the U.S. The conversation surrounding the immigrant experience in the U.S. has focused on economic outcomes and immigrant health and wellbeing. Our research has shown that socioeconomic outcomes contribute to migrants' perceptions of SRH, emphasizing a more holistic approach to understanding the health and wellbeing of immigrants overall. Future research should continue to examine how the immigrant integration process affects not only migrants' perceptions of their health overtime, but also what aspects of their health matter at different points of integration (e.g., naturalization).

## Supporting information

**S1 Table. Allostatic load components and clinical risk cutoffs.**
(DOCX)

**S2 Table. Weighted means and proportions of covariates by inclusion in the 1-year sample (vs. full sample), health of Philippine Emigrants Study, N = 1637.**
(DOCX)

**S3 Table. Binary logistic regression of baseline good/very good/excellent self-rated health on migrant status, Health of Philippine Emigrants Study (HoPES), N = 1632.**
(DOCX)

**S4 Table. Ordinal logistic regression of self-rated health on migrant status, Health of Philippine Emigrants Study (HoPES), N = 1632.**
(DOCX)

**S5 Table. Binary logistic regression of 1-year good/very good/excellent self-rated health on migrant status, Health of Philippine Emigrants Study (HoPES), N = 1194.**
(DOCX)

**S6 Table. Ordinal logistic regression of 1-year self-rated health on migrant status, Health of Philippine Emigrants Study (HoPES), N = 1194.**
(DOCX)

## Author Contributions

**Conceptualization:** Adrian Matias Bacong, Gilbert C. Gee.

**Data curation:** Adrian Matias Bacong, Anna K. Hing.

**Formal analysis:** Adrian Matias Bacong, Anna K. Hing.

**Funding acquisition:** A. B. de Castro, Gilbert C. Gee.

**Investigation:** Adrian Matias Bacong.

**Methodology:** Adrian Matias Bacong, Catherine M. Crespi.

**Project administration:** A. B. de Castro, Gilbert C. Gee.

**Resources:** Gilbert C. Gee.

**Software:** Adrian Matias Bacong.

**Supervision:** Brittany Morey, A. B. de Castro, Gilbert C. Gee.

**Writing – original draft:** Adrian Matias Bacong, Anna K. Hing, Brittany Morey, A. B. de Castro, Gilbert C. Gee.

**Writing – review & editing:** Adrian Matias Bacong, Anna K. Hing, Brittany Morey, Catherine M. Crespi, Maria Midea Kabamalan, Nanette R. Lee, May C. Wang, A. B. de Castro, Gilbert C. Gee.

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
