## [Decision Letter · Decision Letter 0]

28 Jan 2022

PGPH-D-21-00506

Health Selection on Self-Rated Health and the Healthy Migrant Effect: Baseline and 1-Year Results from the Health of Philippine Emigrants Study

Dear Adrian Matias Bacong

Thank you for submitting your manuscript to PLOS Global Public Health. After careful consideration, we feel that it has merit but does not fully meet PLOS Global Public Health’s publication criteria as it currently stands. Therefore, we invite you to submit a revised version of the manuscript that addresses the points raised during the review process.

Please find reviewers' comments below. Minor revisions are requested as outlined. I look forward to receiving your revised manuscript.

We look forward to receiving your revised manuscript.

Kind regards,

Jo Vearey

Academic Editor

Journal Requirements:

1. Please amend your detailed Financial Disclosure statement. This is published with the article, therefore should be completed in full sentences and contain the exact wording you wish to be published.

Please state what role the funders took in the study. If the funders had no role in your study, please state: “The funders had no role in study design, data collection and analysis, decision to publish, or preparation of the manuscript.”

2. Please update your Competing Interests statement. If you have no competing interests to declare, please state: “The authors have declared that no competing interests exist.”

3. In the online submission form, you indicated that “The data underlying this article will be shared on reasonable request to the corresponding author.”. All PLOS journals now require all data underlying the findings described in their manuscript to be freely available to other researchers, either 1. In a public repository, 2. Within the manuscript itself, or 3. Uploaded as supplementary information.

4. We notice that your supplementary tables are included in the manuscript file. Please remove them and upload them with the file type 'Supporting Information'. Please ensure that all Supporting Information files are included correctly and that each one has a legend listed in the manuscript after the references list. 

Additional Editor Comments (if provided):

Dear authors

Thank you for the opportunity to review your paper. Please find reviewer comments attached.

I look forward to receiving your revised manuscript.

Warm wishes

Jo

Reviewers' comments:

Reviewer's Responses to Questions

**Comments to the Author**

1. Does this manuscript meet PLOS Global Public Health’s publication criteria? Is the manuscript technically sound, and do the data support the conclusions? The manuscript must describe methodologically and ethically rigorous research with conclusions that are appropriately drawn based on the data presented.

Reviewer #1: Yes

Reviewer #2: Yes

2. Has the statistical analysis been performed appropriately and rigorously?

Reviewer #1: Yes

Reviewer #2: Yes

3. Have the authors made all data underlying the findings in their manuscript fully available (please refer to the Data Availability Statement at the start of the manuscript PDF file)?

Reviewer #1: Yes

Reviewer #2: Yes

4. Is the manuscript presented in an intelligible fashion and written in standard English?

Reviewer #1: Yes

Reviewer #2: Yes

5. Review Comments to the Author

Reviewer #1: Thank you for the opportunity of reading this interesting manuscript. The study is original and relevant, it is focused on one particular migrant group (emigrant Philippines in the US) but provides wider insights on the health selection model. I think the manuscript is well-written and adequate for publication, but could be improved for including the following comments/corrections:

1. In intro, when mentioning allostatic load concept, it should be defined and referrenced.

2. In intro, page 6 the paragraph that reads "Given the abundance of mixed evidence..." I actually found only 5 papers related to this, I would not acll it abundant... Please specify more clearly and consistent with existing data.

3. In intro, page 8, Filipino migrants are described as a unique type of migrant, how do they compare to other migrants relevant to the US and the world... Could you better justify this, in order to improve the relevance of the paper?

4. In intro page 9 right before methods, you describe 2 hypothesis, which seem correct but assume independence between dimensions or variables... I am a bit worried about this, as they are interconnected. Could you better justify this, as well as your general OLS approach versus other techniques like SEM in relation to your objectives?

5. Methoids: the study groups are people who legally migrated from the Philippes. Is there undocumented Filippino migration to the US? how large is it? How this study would/not inform abiout them?

6. Age range of study participants 20-59, why? many migrants, as explained in the paper, come for reunification and could be older...

7. Methods please better specify how stratified random sampling of non-migrant people was conducted. Why these cities only? do most migrants of the study came from these cities? (you only list refs 24.25, please give some more detail)

8. Methids many variables are instruments, no discussion around intercultural adequacy for Filippinos was described, please add. How stable are these measures over time in similar populations?

9. The healthy diet variable seems really rough, please justify.

10. In socioeconomic factors you added healthcare utilization, which is often treated separately. Why? Would it not potentially affect self-rated health of migrants over time (beyond first year....)

11. Discussion: in the best model, 44% of variace is explained by the model. This is highlighted in the paper, but little is discussed around the other 66% that needs further explanation. Please expand.

12. Discussion mentions the "health in all policies approach" but it is very poorly explained and developed. Please expand in light of your findings.

Reviewer #2: Self-rated health (SRH) is an important subject for consideration in migration and health research. The authors of this manuscript have authoritatively stated this in the Introduction. Interestingly, the authors have pointed out that there are factors as determinants responsible for this situation that needs to be clearly defined and their dynamics understood. Thus, the study is set to examine differences in self-rated health between migrants to the U.S. and a comparable group of non-migrants at baseline (premigration) and one year later, accounting for seven domains: physical health, mental health, health behavior, demographics, socioeconomic factors, psychosocial factors, and social desirability. This study design would contribute to the existing literature on migration and health as it allows us to compare migrants to those who did not migrate in the country of origin (i.e. non-migrants). It also allows us to examine health selection at two time points: pre-migration and 1-year post-migration.

In the context of social determinants of (migration) health which the study seems to refer, this reviewer has these observations and comments;

1. With this as context, the reviewer suggests an inclusion of social determinants of (migration) health frame in the literature review. This would make the introduction clearer and connect well with the study.

2. The absence of a review of theoretical constructs of social determinants of health and how this links with health and impact on migration has weakened the conceptual grounding of the study.

3. Methodological approach would be strengthened by stating the study design first, followed by a statement showing, in clear terms as the convention is, that the study is part of a national study.

4. The statistical analysis appears to be properly done and interpretation of the results is correct. An improvement could be done on being consistent with the use of conventional statistical symbols (b, B and beta).

5. Discussion of the results could be more structured to highlight more on the limitations of the methodology employed considering that this was only at two time points.

Overall, this is an interesting paper to read.

6. PLOS authors have the option to publish the peer review history of their article (what does this mean?). If published, this will include your full peer review and any attached files.

**Do you want your identity to be public for this peer review?** For information about this choice, including consent withdrawal, please see our Privacy Policy.

Reviewer #1: No

Reviewer #2: **Yes: **Mphatso Kamndaya

---

## [Editor Report · Decision Letter 1]

25 Apr 2022

PGPH-D-21-00506R1

Health Selection on Self-Rated Health and the Healthy Migrant Effect: Baseline and 1-Year Results from the Health of Philippine Emigrants Study

Dear Dr. Bacong,

Thank you for submitting your manuscript to PLOS Global Public Health. After careful consideration, we feel that it has merit but does not fully meet PLOS Global Public Health’s publication criteria as it currently stands. Therefore, we invite you to submit a revised version of the manuscript that addresses the points raised during the review process.

We look forward to receiving your revised manuscript.

Kind regards,

Biplab Kumar Datta, Ph.D.

Academic Editor

Journal Requirements:

Additional Editor Comments (if provided):

Thank you for addressing the comments made by the reviewers. However, there remains some technical issues that needs to be clarified before this paper can be recommended for publication.

1. First, the measure of SRH was regarded continuous in the analyses. The results of the OLS (in Table 2) suggest that the expected level of SRH among migrants is 0.77 unit higher than that among non-migrants. However, the SRH status is subjective; and movement from poor to fair (i.e., 0 to 1) and very good to excellent (i.e., 3 to 4) though by 1 units in both cases but very different in subjective measure. Therefore, it warrants solid rationale why a continuous measure of SRH is appropriate to answer the question asked in this paper. Please justify the use of continuous SRH measure as the outcome variable and adequately discuss the statistical issues related to it. Also please adequately discuss the interpretation of a continuous SRH in the context of the research question, and clearly outline any limitations associated with the choice of outcome variable.

2. Second, one of the necessary criteria of ordinal logistic regression in fulfillment of the proportional odds assumption. If that assumption does not hold, then the estimates of a ordinal/ordered logistic regression is not valid. I do not not any mention of the proportional odds assumption in the manuscript, satisfying which is a prerequisite of carrying out an ordinal logistic specification. Please conduct appropriate statistical checks to confirm whether the proportional odds assumption was duly satisfied. If yes, clearly mention that in the manuscript. If no, then replace the ordinal logistic analysis with multinomial logistic model and report the results accordingly.

3. It is not explicitly mentioned how standard errors were estimated in the OLS and the logistic regression framework. I guess, the authors assumed homoskedasticity as I did not see any mention of robust standard errors. Homoscedasticity or constant variance of the error term is rare in real life data, and it is convention to provide robust standard error estimates. Please clearly state how the standard errors were estimated in the regression. If it is not robust, provide evidence of homoscedasticity. If the errors are heteroskedastic, then provide appropriate standard errors (or confidence intervals) that are robust to heteroskedasticity.

I will be looking forward receiving your takes on these technicalities and the revised manuscript.
---

## [Editor Report · Decision Letter 2]

4 Jun 2022

Health Selection on Self-Rated Health and the Healthy Migrant Effect: Baseline and 1-Year Results from the Health of Philippine Emigrants Study

PGPH-D-21-00506R2

Dear Dr. Bacong,

We are pleased to inform you that your manuscript 'Health Selection on Self-Rated Health and the Healthy Migrant Effect: Baseline and 1-Year Results from the Health of Philippine Emigrants Study' has been provisionally accepted for publication in PLOS Global Public Health.

Best regards,

Biplab Kumar Datta, Ph.D.

Academic Editor